# Ergodicity-breaking reveals time optimal decision making in humans

**David Meder** [1], **Finn Rabe** [1,2], **Tobias Morville** [1], **Kristoffer H. Madsen** [1,3], **Magnus T. Koudahl** [1,4], **Ray J. Dolan** [5], **Hartwig R. Siebner** [1,6,7], **Oliver J. Hulme** [1] *

**1** Danish Research Centre for Magnetic Resonance, Copenhagen University Hospital Amager and Hvidovre, Copenhagen, Denmark, **2** Neural Control of Movement Lab, ETH Zurich, Zurich, Switzerland, **3** Department of Applied Mathematics and Computer Science, Technical University of Denmark, Kongens Lyngby, Denmark, **4** Department of Electrical Engineering, Eindhoven University of Technology, Eindhoven, the Netherlands, **5** Max Planck UCL Centre for Computational Psychiatry and Ageing Research, London, United Kingdom, **6** Department of Neurology, Copenhagen University Hospital Bispebjerg, Copenhagen, Denmark, **7** Institute for Clinical Medicine, University of Copenhagen, Copenhagen, Denmark

* oliverh@drcmr.dk

**Data Availability Statement:** The datasets, analyses, stimuli, code, and codebook are available in the 'ergodicity-breaking-choice-experiment" repository: github.com/ollie-hulme/ergodicity-breaking-choice-experiment. All data figures have

## Abstract

Ergodicity describes an equivalence between the expectation value and the time average of observables. Applied to human behaviour, ergodic theories of decision-making reveal how individuals should tolerate risk in different environments. To optimize wealth over time, agents should adapt their utility function according to the dynamical setting they face. Linear utility is optimal for additive dynamics, whereas logarithmic utility is optimal for multiplicative dynamics. Whether humans approximate time optimal behavior across different dynamics is unknown. Here we compare the effects of additive versus multiplicative gamble dynamics on risky choice. We show that utility functions are modulated by gamble dynamics in ways not explained by prevailing decision theories. Instead, as predicted by time optimality, risk aversion increases under multiplicative dynamics, distributing close to the values that maximize the time average growth of in-game wealth. We suggest that our findings motivate a need for explicitly grounding theories of decision-making on ergodic considerations.

## Author summary

How people take risks is central to our understanding of how they make decisions. Theories of decision making commonly assume that preferences for risk are like personality traits, being both idiosyncratic to individuals and stable over time. A new theory based on the thermodynamic concept of ergodicity predicts that risk preferences should be determined by the dynamical settings that people make decisions in. We show that a simple manipulation of the dynamics of a gambling game exerts a strong and systematic effect on people's willingness to take risks. The level of risk taking and how this changed with different dynamics was quantitatively predicted from first principles within ergodic theory. We show that existing theories of decision making cannot adequately account for these changes in risk preference. This work is relevant across the behavioral sciences insofar as

associated raw data. There are no restrictions on data availability.

**Funding:** HRS received grants from the Novo Nordisk Foundation: (NNF14OC0011413) and the Lundbeck Foundation (R59 A5399 and R186-2015-2138). OJH received grants from the Lundbeck Foundation (R140-2013-13057), and from the Danish Research Council (12-126925). DM received a grant from the Novo Nordisk Foundation (NNF16OC0023090). The funders had no role in study design, data collection and analysis, decision to publish, or preparation of the manuscript.

**Competing interests:** I have read the journal's policy and the authors of this manuscript have the following competing interests: HRS has received an honoraria as speaker from Sanofi Genzyme (Denmark) and Novartis (Denmark) and as a consultant from Sanofi Genzyme (Denmark), and as a senior editor (NeuroImage) from Elsevier Publishers (Amsterdam, The Netherlands). HRS has received royalties as a book editor from Springer Publishers (Stuttgart, Germany).

it challenges the validity of one of the most widespread assumptions in modern decision theory.

## Introduction

Ergodicity is a foundational concept in models of physical systems that include elements of randomness[1,2]. A physical observable is said to be ergodic if the average over its possible states, is the same as its average over time. For instance, the velocity of gas molecules in a chamber is ergodic if averaging over all molecules at a fixed time (an expectation value) yields the same value, as averaging a single molecule over an extended period (a time average). In other words, ergodicity ensures an equality between the time average and the expectation value. The relevance of ergodicity to human behavior is that it provides important constraints for thinking about how agents should compute averages when making decisions[3].

In the behavioral sciences, decision making is studied predominantly using experiments with additive dynamics, where choice outcomes exert additive effects on wealth. An agent might gamble on a coin toss for a gain of $1 each time they win, they might score a point each time they correctly execute a motor action, and so on. In these examples, changes in wealth are ergodic, and in such settings a linear utility function is optimal for maximising the growth of wealth over time[3,4]. In other words, for this utility function, when changes in expected utility are maximized per unit time, this maximizes the time average growth rate of wealth (Fig 1F). The time average growth rate of wealth under an additive setting is calculated simply as the additive change in wealth per unit time (Eq 4 in Methods), either over a finite or infinite time horizon.

However, not all dynamics that individuals face are additive. Some dynamics in the environment are multiplicative, for instance. Examples of multiplicative dynamics include stock market investments, compound interest on savings, and the spread of infectious diseases. The time average growth rate of wealth under a multiplicative dynamic is calculated as the exponential growth of wealth per unit time (Eq 5 in Methods). Settings with multiplicative wealth dynamics have non-ergodic wealth changes, which means that the expectation value of changes in wealth no longer reflects the time-average growth. Indeed, there are gambles in which changes in wealth have a positive expectation value, but have a negative time average growth rate[4]. A simple example is a fair coin gamble: heads to gain 50% of one's current wealth, tails to lose 40% of one's current wealth. Counterintuitively, whilst this gamble has a positive expectation value (where wealth grows by a factor of 1.05 per trial: [1.5 + 0.6]/2), it has a negative time average growth rate (where wealth grows by a factor of ~0.95 per trial: sqrt [1.5*0.6]). Whilst a full explanation of this discrepancy requires invoking the behavior of different types of limit, a more intuitive explanation is that at any one time, the majority of players will have experienced negative time average growth, but a minority of players will experience such extreme wealth growth that this dominates the expectation value. For this gamble however, maximizing expected value eventually leads to ruin for all players. In such multiplicative settings a logarithmic utility function is time optimal, since maximizing changes in expected utility per unit time then maximizes the time average growth rate of wealth[3] (Fig 1G). The reason is trivial in the sense that the time average growth rate is calculated as the average change in logarithmic wealth per unit time.

These examples highlight the fact that time optimal behavior relies on agents adapting their utility functions according to the dynamics of their environments. Time optimality here refers to the optimality of a behavioral strategy in maximising the time average growth rate of wealth.

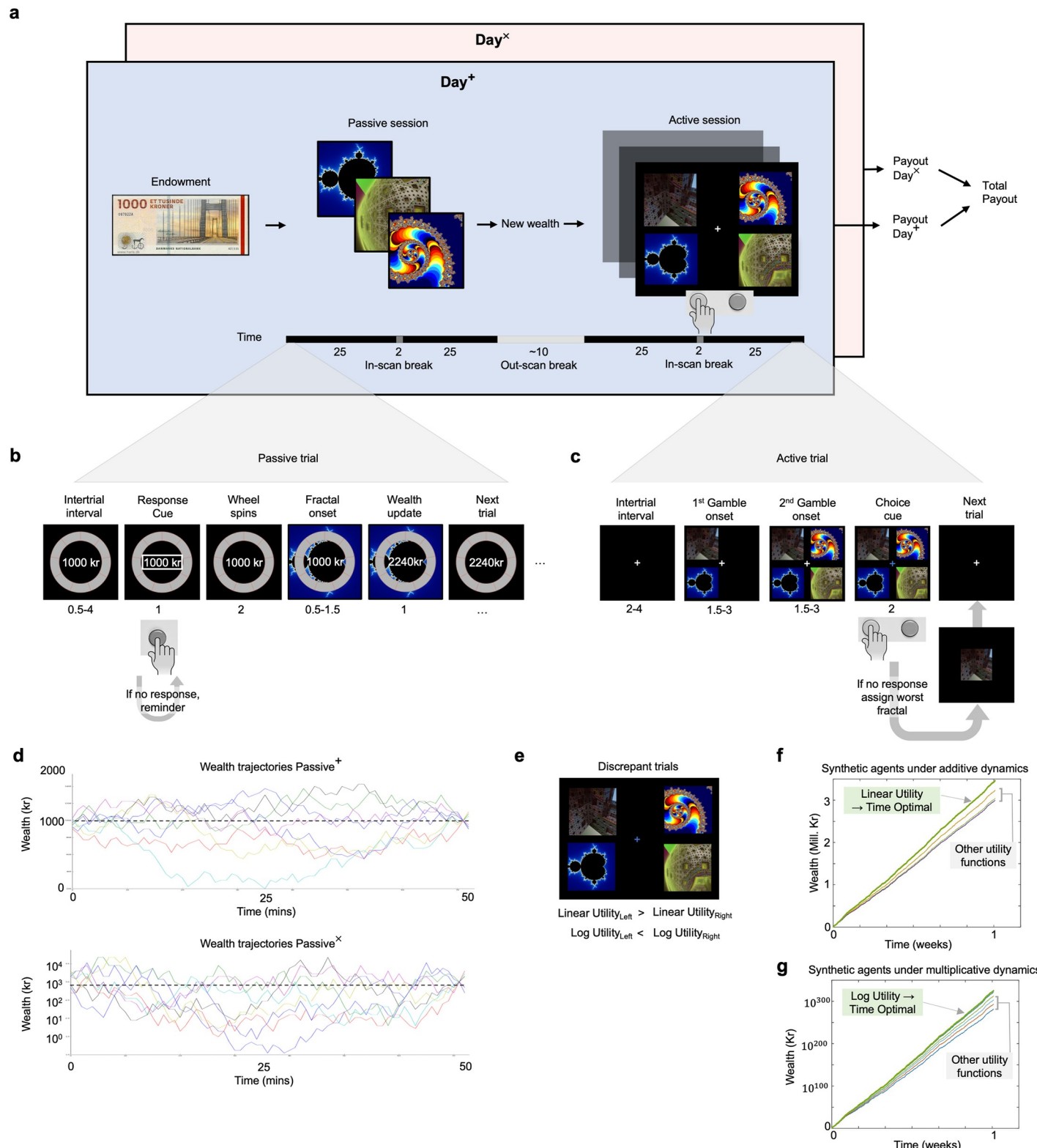

**Fig 1. Experimental design and wealth trajectories. A**, two-sheets (blue and pink) summarise the repeated protocol for both days, which only differ in the dynamics of wealth changes. Numbers indicate durations in minutes. Although three stimuli are shown for illustration, a total of 9 stimuli were used in each session. **B**, single trial from a passive session, where durations are in seconds and ranges depict a uniformly distributed temporal jitter. **C**, single trial from an active session. **D**, wealth trajectories in real time over the course of each passive session. The trajectory for Passive$^{\times}$ is plotted on a log scale, appropriate to the multiplicative dynamics. Eight

randomly selected trajectories are plotted. Dotted line shows initial endowment level of 1000DKK. **E**, discrepant trials are a subset of trials, where agents with linear and logarithmic utility functions would be predicted to make different choices. In the example here, an agent with linear utility would choose the left-hand gamble whereas an agent with logarithmic utility would choose the right-hand gamble. **F,** wealth trajectories of synthetic agents with different utility functions (prospect theory and isoelastic) repeatedly playing the set of additive gambles over one week (For details see S2 Text). The agent with linear utility has the highest time average growth rate (green). **G,** equivalent simulations for multiplicative gambles. The agent with log utility has the highest time average growth rate (green). The time optimal agent is an agent with linear utility for additive dynamics, and log utility for multiplicative dynamics, and thus also experiences both wealth trajectories depicted in green (in F and G). We note that the time optimality of an agent's behavior is independent of the timescale, even if the consequences of time optimality may require a particular timescale to become visible in noisy plots. In F and G, the time optimal utility function is visibly advantageous on the timescale of hours.

A strategy or utility function that affords maximising the time average growth rate of wealth is thus said to be time optimal (this is distinct from notions of optimality which refer only to the consistency of choices). In contrast to time optimality, prevailing formulations of utility theory, including expected utility theory[5,6] and prospect theory[7,8], are not premised on the dynamics of the environment. In treating all possible dynamics as the same, these formulations imply that utility functions are indifferent to the dynamics. Since standard decision theories assume stable but idiosyncratic utility functions, whereas time optimality prescribes specific utility functions for specific dynamics, the two classes of theory make substantively different predictions. Here we manipulated the ergodic properties of a simple gambling environment, by switching between gambling for additive increments of money versus gambling for multiplicative growth factors, evaluating the effect this has on the utility functions that best account for choices. We found evidence that gamble dynamics impose a consistent effect on utility functions, with participants shifting from approximately linear utility functions in the additive session, to approximately logarithmic utility functions in the multiplicative session. This effect was better predicted by a time optimal model compared to mainstream utility models.

## Methods

### Ethics statement

Informed written consent was obtained from all subjects as approved by the Regional Ethics Committee of Region Hovedstaden (protocol H-17006970) and in accordance with the declaration of Helsinki.

### Methods summary

We asked whether switching between additive and multiplicative gamble dynamics systematically influences decision making under risk. Specifically our objective is to investigate how existing utility models—primarily prospect theory and isoelastic utility[9]—perform in comparison to a null model of time optimality in explaining choice behaviour under different dynamics. In an experiment that spanned two separate days, each subject engaged in a gambling paradigm with either additive or multiplicative dynamics in their in-game wealth (dynamics, hence). At the start of each day, participants were endowed with an initial wealth of 1000DKK / ~$155 (Fig 1A), after which they took part in a passive session during which they had an opportunity to learn, via observation, the deterministic effect of image stimuli on their endowed wealth (Fig 1B). On the additive day ($Day^+$) the stimuli caused additive changes in wealth whereas on the multiplicative day ($Day^\times$) the stimuli caused multiplicative changes to their wealth (Eqs 1–5, S1 Fig in S1 Text). Different stimuli were used for the two different days and the association between the stimuli and the change in wealth was randomized between subjects. Having repeatedly observed these contingencies between the stimuli and the changes in wealth, subjects subsequently engaged in an active session during which they repeatedly chose between two gambles composed of pairs drawn from the same set of stimuli

(Fig 1C, Eqs 6–9). Upon choosing a gamble, each of the two stimuli had a 50% probability of being the outcome of the gamble. Subjects understood that the gamble outcomes were not revealed during the game, and that 10 of the outcomes of the chosen gambles would be randomly realised at the end of each day and applied to their current in-game wealth. The dynamics exist in the active session insofar as the fractals impose dynamical changes in wealth when they are realised at the end of the experiment. There were four sessions in total per subject, Passive$^\times$ and Active$^\times$ occurring on Day$^\times$; and Passive$^+$ and Active$^+$ occurring on Day$^+$. We adopted three complementary analysis strategies: The first is model-independent in the sense that we tested whether choice frequencies change according to different dynamics. The second and third approaches were model-dependent insofar as we formally compare theoretical models of utility in terms of their parameter estimates, and in terms of the predictive adequacy of each utility model.

## Subjects and power

This paper focuses on the behavioral data obtained from a neuroimaging study on the neural encoding of utility. The criteria for inclusion were being aged 18–50 and fluent in English. The criteria for exclusion were: a history of psychiatric or neurological disorder, credit problems (operationalized via bad pay status on www.dininfo.dk), or expertise in a quantitative or cognitive domain (finance, banking, accountancy, economics, mathematical sciences, computer science, engineering, physics, psychology, neuroscience). Neuroimaging specific exclusion criteria were also applied, including implanted metallic or electronic objects, heart or brain surgery, severe claustrophobia, or inability to fit into the scanner (weight limit of ~150kg, bore diameter of 60 cm). Except for the latter, all such information was self-reported. The intended sample size was 20, however due to post-hoc exclusion (1 participant fell asleep, 1 failed to learn the stimuli) the achieved sample size was 18 (6 females, age: M = 25.79, SD = 4.69, range 20–38). Subjects were recruited as a convenience sample, via the subject recruitment website www.forsøgsperson.dk. The sample number was based on general guidelines for the minimal number of subjects required for medium effect sizes in neuroimaging datasets[10]. The number, timing, and jittering (randomized timing) of events within each session was based on prior efficiency simulations for similar neuroimaging paradigms. As such, no a priori design analyses were performed for the behavioral data only. No stopping rule or interim analyses were performed. Data collection ran from the 10/06/2017 to 30/07/2017. All data was acquired at the Danish Research Centre for Magnetic Resonance. Independent of their payouts in the gambling paradigm, all subjects were compensated 1020 DKK / ~$160 for a grand total of 6 hours of participation over the two days. A forthcoming paper will focus primarily on the neuroimaging data.

## Experimental procedure

After changing into hospital gowns, subjects were read the instruction sheet (see below and S6 Text). To précis, subjects were truthfully informed that the aim of the experiment was to study how the brain reacts to changes in wealth, that all the money involved is real, and that the total accumulated wealth will be paid out as the sum of that accumulated over the two days (Fig 1A). They then played ~20 demo trials of the paradigm in the scanner control room, including both active and passive sessions (~5mins) for no financial consequence. The experimenter demonstrated what happened if buttons were not pressed in time (Fig 1B and 1C). Subjects were instructed that each day lasts 3 hours in total, with ~60mins for the passive session (inc. time for localiser scan and shim), a short break, then ~60-75mins for the active session (inc. localiser, shim, anatomical scans), with short breaks within the session (Fig 1A). Each subject

entered the scanner and was set up with a respiratory belt to monitor breathing and with a pulse meter on the middle or index finger of the non-responding hand. All stimuli were projected under dark conditions onto a screen located within the bore of the scanner (Siemens, MAGNETOM Prisma), and viewed via mirrors mounted to the head coil. Subjects were instructed to always fixate the central fixation cross (Fig 1B and 1C) and choose via button box. The paradigm was presented via the Psychopy2 toolbox (v1,84.2) running on Python (2.7.11).

## Experimental design

The experiment is a fully crossed randomized controlled trial in which the wealth dynamic is the primary independent variable, and choice in the active task is the primary observable. The wealth dynamic, as well as the deterministic association between stimuli and outcomes was controlled via a computer programme and thus double blinded. Further, since payouts at the end of the test day were subject to being randomly realized from each subject's choices as well as being statistically balanced between conditions, payout was also effectively double-blinded. Subjects were neither informed of any explicit details concerning dynamics or differences between test days, nor given any reason to expect that the test days were different. The instructions, procedures, and setup were otherwise identical for both test days. The order in which multiplicative and additive test days were conducted was randomized and counterbalanced across the group. Subjects were not able to make notes or use a calculator due to their location inside the brain scanner. Measures collected but not included in this report include all functional and structural neuroimaging modalities, physiological noise measurements (pulse rate and breathing), and reaction times. To ensure good quality model estimation, we recorded many decisions (312 in total per active session) spanning a large subspace (144) of the possible unique gamble combinations. To avoid the problems associated with gambling for "peanuts"[11], the outcomes of decisions are for large quantities of money on each trial (mean possible change in wealth $Day^{\times}$ = 413.07 DKK / per decision, SD = 249.78, range = -422.87 to 946.71, mean possible change in wealth $Day^{+}$ = 267.76 DKK per decision, SD = 119.20, range = -428–428). Subjects were thus strongly incentivized to pay attention to all the stimuli and to optimize their decision-making throughout the active sessions.

## Pre-registration and deviations

The experimental protocol was preregistered at www.osf.io/9yhau. There was one deviation from the protocol: The preregistration stated that in the $Passive^{+}$ session, the final additional stimulus applied to their wealth after having returned to 1000DKK (see section "Passive session stimulus sequences" below) would exclude the most extreme stimuli. Those were however included in the paradigm.

## Passive session instructions

Subjects were instructed in English as follows: *"For the passive phase, you will see a number in the middle of the screen, this is your current wealth for the day in kr. When you see a white box around the number, you are to press the button within 1s. (If you do not, you will be instructed to "press button earlier"). Shortly after pressing the button you will see an image in the background, and this will cause your wealth to change. You are instructed to attend to any relationship between the images and the effect this has on your wealth, since in the active phase that follows you will be given the opportunity to choose images to influence your wealth. Learning these relationships can make a large difference to your earnings in the active phase."* These instructions were identical on both days to avoid biasing the subject toward any particular strategy. Full subject instructions are provided in S6 Text.

## Passive session dynamics

Formally the passive session can be described as follows: At the start of each test day, subjects were endowed with an initial wealth $x(t_0)$ of 1000DKK, which defined their wealth at the first timepoint, which we denote as $t_0$. Independently for each subject, 9 stimuli were randomly assigned (from a fixed set of 18) for Day$^+$, with the remaining 9 assigned to Day$^\times$. Each stimulus, viewed at time $t$ was programmed to have a deterministic effect on the subject's wealth $x(t)$, with the sequence of stimuli causing stochastic fluctuations in wealth (Fig 1D). The sequence of stimuli deterministically caused dynamics in their wealth which can be expressed as:

$$x(t + \delta t) = x(t) \circledast s(t), \tag{1}$$

where $\circledast$ is a wildcard operator, which on Day$^+$ is the addition operator $+$, and on Day$^\times$ is the multiplication operator $\times$. $s(t)$ is a random outcome variable drawn from set $S^\times$ on Day$^\times$, and from set $S^+$ on Day$^+$ (see S1A Fig in S1 Text). This means that the type of wealth dynamic that the stimuli caused was determined by the test day. On Day$^\times$ under multiplicative dynamics, the outcome $s(t)$ is the realisation of a random multiplier (growth factors) that can range from ~doubling at one extreme, to ~halving at the other (equally spaced on a logarithmic scale). On Day$^+$, under additive dynamics, the outcomes $s(t)$ is the realisation of a random increment, ranging from +428 to -428DKK (equally spaced on a linear scale). Though the dynamics are qualitatively different, to constrain the differences in wealth changes between conditions, we set the bounds of the random increments for Passive$^+$ to the central 85th percentile interval of the absolute wealth changes on Day$^\times$.

## Passive session stimulus sequences

The stimulus sequence was randomized such that wealth levels were constrained to lie in the interval (0 $kr$, 5000 $kr$) at all times. This was achieved by presenting each of the 9 stimuli 37 times (and the ensuing effect on wealth), thus generating a set of 333 stimuli. The sequence order was randomized without replacement. Any sequence that resulted in a partial sum larger than 5000 or lower than 0DKK, would be rejected and another random sequence generated. This was necessary to render the experiment subjectively plausible, and to avoid debts which for ethical reasons could not be realised. Since each stimulus was presented with equal frequency, at the end of these 333 trials in the additive condition, the finite time average additive growth rate was zero kr per unit time. Equivalently, at the end of the 333 trials in the multiplicative condition, the finite time average multiplicative growth rate amounted to a growth factor of one per unit time. Thus, at the end of these 333 trials, in both conditions subjects had returned to their initial endowed wealth of 1000 DKK. One additional stimulus was then shown and applied to their wealth, meaning that all subjects had a randomly determined wealth level, as they had been informed (Fig 1D).

## Passive session wealth trajectories and growth

The wealth at the end of the Passive$^+$ session can be calculated as:

$$x(t_0 + T\delta t) = x(t_0) + \sum_{\tau=1}^{T} s(\tau), \tag{2}$$

and for the Passive$^\times$ session as:

$$x(t_0 + T\delta t) = x(t_0) \prod_{\tau=1}^{T} s(\tau), \tag{3}$$

where, in both equations, $s(\tau)$ is the random outcome variable in round $\tau$, and T is the total

number of trials in the passive sessions. The finite time average growth of wealth on Day$^+$ can be calculated as:

$$\bar{g}_{\Delta t}^{+} = \frac{\Delta x}{\Delta t},$$

(4)

where $\Delta x = x(t_0+T\delta t)-x(t_0)$, and $\Delta t = T\delta t$. On Day$^\times$ this is calculated as:

$$\bar{g}_{\Delta t}^{\times} = \frac{\Delta \ln x}{\Delta t}.$$

(5)

This design ensured substantial opportunity for subjects to learn the causal effects of each stimulus, whilst also not accumulating extremely high or low wealth levels.

## Active session instructions

After the passive session, the subjects had a short break of ~5mins outside of the scanner before returning to engage in an active choice task in which they repeatedly decided between two different gambles composed of the stimuli they had just learnt about (Fig 1A). Subjects were instructed as follows: *"With the money accumulated in the passive phase, you will play gambles composed of the same images. In each trial, you will be presented with two of the images that you have learned about in the passive phase. By pressing the buttons in the scanner to move a cursor, you now have the option to choose to either: a) Accept gamble one, in which case you will be assigned one of the two images, each with 50% probability (not shown), or... b) Accept gamble two, in which case you will be assigned one of the two images, each with 50% probability (again not shown). The outcomes of your gambles will be hidden from you, and only 10 of them will be randomly chosen and applied to your current wealth. You will be informed of your new wealth at the end of the active phase. You can keep any money accumulated after the active phase. If you do not choose in time, then we will give you one of the worst images, it is recommended that you always choose in time."* These instructions were identical on both days to avoid biasing the subject toward any particular strategy.

## Active session gambles

As shown in Fig 1C, within a trial, subjects first saw the first gamble of a pair of gambles. This gamble is composed of two stimuli on the left-hand side of the screen, each of which they knew has a 50% chance of being applied to their wealth should this gamble be chosen and realised at the end of the day. We refer to this as the left gamble, $Q^{(Left)}$. 1.5–3 seconds later (uniformly distributed), on the right they saw another two stimuli, here comprising the right gamble $Q^{(Right)}$. In a two alternative forced choice, on each trial, subjects choose via button press between gamble $Q^{(Left)}$ and $Q^{(Right)}$. Formally the gambles are:

$$Q^{(Left)} \bigg| = \begin{cases} s_1^{(Left)}, p_1^{(Left)} = 0.5 \\ s_2^{(Left)}, p_2^{(Left)} = 0.5 \end{cases}$$

(6)

$$Q^{(Right)} \bigg| = \begin{cases} s_1^{(Right)}, p_1^{(Right)} = 0.5 \\ s_2^{(Right)}, p_2^{(Right)} = 0.5 \end{cases}$$

(7)

Choosing between two gambles eliminates any confounds caused by potential preferences for or against gambling. Note that all probabilities are equal and correspond to a fair coin, such that these are easily communicated and to control for any probability distortion effects. The outcome of each gamble was hidden from subjects to avoid subjects being "conditioned" to prefer stimuli as a function of the stochastic pattern of previous outcomes. This also

prevents mental accounting, where subjects keep track of what they have earnt, which introduces idiosyncratic path dependencies between subjects.

## Active session growth rates

For any gamble we can calculate its time average growth rate. The time average additive growth rate for the left-hand gamble is:

$$\bar{g}^{+(Left)} = \left\langle \frac{s^{(Left)}}{\delta t} \right\rangle, \tag{8}$$

and equivalently for the right-hand gamble. The time average multiplicative growth rate for the left-hand gamble is:

$$\bar{g}^{\times(Left)} = \left\langle \frac{\ln s^{(Left)}}{\delta t} \right\rangle, \tag{9}$$

and equivalently for the right-hand gamble. Note that the angled brackets indicate the expectation value operator. Also note that since there were no numerical or symbolic cues at this point, any decision could only be based on their memory of each stimulus (Fig 1C).

## Active session gamble space

On any test day, for any one gamble, there are 81 possible combinations of stimuli ($9^2$, see S1B Fig in S1 Text), and 6561 possible pairs of gambles ($81^2$). This gamble-choice space is too large to exhaustively sample, and contains many gambles that do not discriminate between our hypotheses, and thus we imposed the following constraints: All gambles should be mixed (composed of a gain and a loss), and no two stimuli presented in one trial should be the same, this reduces the gamble choice space down to 144 unique non-dominated choices between gambles: 16 mixed gambles (red text cells, in S1B Fig in S1 Text), paired with 9 other mixed gambles with unique stimuli, gives 16*9 possible gamble pairs. Each of these choices was presented twice, resulting in 288 in total. This restriction of the gamble space thus provides a more efficient means of testing the competing hypotheses of this experiment. Subjects were also presented with 24 No-brainer choices, in which both gambles shared an identical stimulus, but differed in a second. These are otherwise known as statewise dominated choices. In these No-brainer choices, the subject should choose whichever gamble includes the better unique stimulus. This offers a direct means of testing whether subjects could accurately rank the stimuli. One participant (#5) failed to choose statewise dominated gambles with a probability > 0.5 and was excluded from further analysis (S4E Fig in S1 Text). All choices were presented in a random order without replacement.

## Subject payout

Subjects were informed of the following on the first test day prior to the passive session: *"At the end of the two days: Your accumulated wealth will be added over the two days, and transferred to your account, within approximately two weeks, and is taxable under standard regulations (B-income). Total earnings = (Wealth after day 1) + (Wealth after day 2). This will be paid over and above your remuneration for participating in the experiment."* Payout on each test day was limited to the range of 0 to 2000DKK for each day, and thus the range of possible grand total payouts was 0 to 4000DKK (excluding compensation for time). Ten gambles were randomly chosen to be realised at the end of each active session. The realised gambles were applied to the subject's wealth in the order they were chosen, however we note that the exact

order makes no difference to the final wealth for a given test day. The order in which gambles are realised would only make a difference to the behavior of the subject (and thus final wealth), if gambles were realised and communicated to the subject within the active session, whilst decisions were still being made. The number of gamble realisations (10) was designed to be large enough that it felt like the choices in the active session counted substantively toward the subject's final wealth, and not so large that the variance was too large exceeding our payment limits.

## Models

### Model summary

The aim of the modelling was to perform both parameter estimation and model selection. All models deployed hierarchical Bayesian methods, estimated via Monte Carlo Markov Chain sampling. For parameter estimation we estimated a hierarchical model of isoelastic utility, whereas for model selection we estimated a hierarchical latent mixture model, to model latent mixtures of three different utility models.

### Model space

Models can be described by specifying three functions: a utility function, a stochastic choice function, and probability-weighting function[12]. Since all probabilities of outcomes are identical in our experiment, we do not deploy any probability-weighting function. The principal objective of the modelling is to compare between different utility functions in accounting for the choice data over both dynamical conditions. We compared three utility models:

**Prospect theory** where changes in value are equal to a power function of changes wealth:

$$\delta u = \begin{cases} (\delta x)^{\alpha_{gain}} & \text{if } \delta x > 0 \\ -\lambda |(\delta x)|^{\alpha_{loss}} & \text{if } \delta x \leq 0 \end{cases}, \tag{10}$$

where $\alpha_{loss}$ and $\alpha_{gain}$ are risk preference parameters lying on the interval (0,1), and $\lambda$ is a loss aversion parameter which lies on the interval $(1, \infty)$, and $\delta x$ is computed as $x(t+\delta t)-x(t)$. $x(t)$ denotes in-game wealth at time t, and $\delta t$ indicates the next time step, which in this game is the next trial ~10 seconds later. Note that, although this is referred to as value within prospect theory itself, it is equivalent to the concept of utility in the other models. We note that some versions of Prospect theory would apply different probability weightings for gain outcomes compared to loss outcomes, however here we assume the same weighting for gains and losses.

**Isoelastic utility** where changes in utility are given by:

$$\delta u = u(x(t + \delta t)) - u(x(t)),$$
$$\text{where}$$
$$u(x(t)) = \frac{x(t)^{1-\eta} - 1}{1 - \eta}, \tag{11}$$

and where $\eta$ is a risk aversion parameter which lies on the real number line, with risk aversion increasing for numbers above 0, and risk seeking increasing for increasingly negative numbers.

**Time optimal utility** is where changes in utility are determined by linear utility under additive dynamics, and by logarithmic utility under multiplicative dynamics:

$$\delta u = \begin{cases} \delta x & \text{if } \text{additive dynamics} \\ \delta \ln(x) & \text{if } \text{multiplicative dynamics} \end{cases} \tag{12}$$

Note that this model follows from one criterion, that agents maximize the time average growth rate of their wealth according to the dynamic they face. These utility functions allow the time average growth rates ($\delta u/\delta t$) under these two dynamics to be computed and maximized by choice.

## Expected utility

For each gamble the expected utility is calculated for each utility model as the expectation value:

$$\langle \delta u^{Left} \rangle = p \cdot \delta u_1^{Left} + p \cdot \delta u_2^{Left}, \tag{13}$$

and equivalently for the right-hand gamble. Differences in utility between the left and right gambles are denoted by $\Delta$ such that the difference in expected utility between the left and right-hand gamble is:

$$\langle \delta u \rangle^{\Delta} = \langle \delta u^{Left} \rangle - \langle \delta u^{Right} \rangle. \tag{14}$$

## Current wealth

It should be noted that the current wealth that enters these three models is stationary over time, fixed at the level obtained from the end of the passive phase. This is because changes in wealth are not realised until the end of the day, which means that all outcomes are hidden from the subject at the time decisions are being made. Whilst it is possible in principle to update one's expected wealth as a function of the decisions already made, this is computationally implausible, especially under the demanding cognitive constraints of the task. To compute expected wealth for a given trial, past choices must be recalled and integrated over all possible outcomes. This integration quickly becomes computationally implausible, especially for the multiplicative condition which must consider all the possible wealth trajectories up to the given point in time (further discussed in S3 Text).

## Stochastic choice function

The stochastic choice function is identical for all models under consideration, and is comprised of a logistic function:

$$\theta\left(\langle \delta u \rangle^{\Delta}\right) = \frac{1}{1 + e^{-\beta \langle \delta u \rangle^{\Delta}}}, \tag{15}$$

where $\beta$ is a sensitivity parameter that determines the sensitivity of the choice probability to differences in the expected change in utility between the two gambles, and where $\theta$ evaluates to the probability of choosing the left-hand gamble. For clarity of presentation, we suppress subscripts and superscripts that denote model, and subject specific parameters. Note that $\beta$ is free to vary over both subjects and conditions for all three models, and thus there are two sensitivity parameters per subject, for each of the three utility models. Allowing the sensitivity parameter to change with the dynamic, allows any potential scaling differences in the change of wealth, to be accommodated in the stochasticity of the choices.

## Sampling procedures

The Bayesian modelling deployed here affords computation of the full probability distributions of parameters, rather than point estimates which ignore the uncertainty with which parameters are estimated. Via its hierarchical structure, individuals are modelled as coming from group-level distributions, such that information from the group informs the estimation of the

individual, and constrains extreme values that might be estimated with uncertainty[13]. To this end, Monte-Carlo Markov Chain sampling was performed via JAGS(v4.03), called from MATLAB (v9.4.0.813654 R2018a, Mathworks®, mathworks.com) via the interface MATJAGS (v1.3, psiexp.ss.uci.edu/research/programs_data/jags). For all models we used: a burn-in $> 500$, $10^4$ samples per chain, and ten chains for model recovery and parameter estimation, and four chains for model selection and parameter recovery. Convergence was established via monitoring R-hat values to fall between 1 and 1.01. These sampling procedures were efficient, as indicated by low autocorrelations of the sample chains, R-hat values, and visual inspections of the chain plots.

## Model selection

The three utility models were estimated via a single hierarchical latent mixture (HLM) model. Whilst these utility models are submodels of the HLM, for consistency we persevere in calling them utility models. The sensitivity parameter $\beta$ parameter is common to all three utility models and is free to vary by subject and by condition, to accommodate any differences in the scaling of wealth changes. Following Nilsson and colleagues[13] we set weakly informative hyperpriors, such that the group mean of $\beta$ was certain to lie in an interval that ranges from 0.1 to ~30. Assuming an uninformative uniform hyperprior distribution for the lognormal group means, this translates to hyperpriors distributed as: $\mu_c^\beta \sim \text{Uniform}(-2.3, 3.4)$, where subscript $c$ denotes experimental condition. We assigned uninformative uniform hyperpriors for the lognormal standard deviations $\sigma_c^\beta \sim \text{Uniform}(0.01, 1.6)$ where 1.6 is the approximate standard deviation of a uniform distribution that ranges from −2.30 to 3.4. *Time optimal utility model*: Specified as a restricted isoelastic model, with a population mean risk aversion $\mu^\eta$ fixed to 0 for additive and 1 for multiplicative dynamics. Assuming uninformative uniform hyperpriors $\sigma_c^\eta \sim \text{Uniform}(0.01, 1.6)$ for the standard deviations of the normally distributed risk aversion parameters means that only the dispersion around the [0,1] coordinate in risk aversion space is free to vary. *Prospect theory utility model*: has three further free parameters. For risk preferences it has one $\alpha$ parameter each for gains and losses, both are constrained to lie between 0 and 1, here assumed to each come from a lognormal distribution, with an uninformative uniform hyperprior distribution on the lognormal group means and standard deviations $\mu^\alpha \sim \text{Uniform}(-2.3, 0)$ and $\sigma^\alpha \sim \text{Uniform}(0, 1.6)$. The third parameter is the loss aversion parameter λ, which we assumed to lie on an interval from 1 and 5, and thus we set equivalent non-informative uniform hyperpriors on the lognormal group means and standard deviations $\mu^\lambda \sim \text{Uniform}(0, 1.6)$ and $\sigma^\lambda \sim \text{Uniform}(0, 1.6)$. *Isoelastic utility model*: Assuming uninformative uniform hyperpriors for the population mean of the risk aversion parameter $\mu^\eta \sim \text{Uniform}(-2.5, 2.5)$ and $\sigma^\eta \sim \text{Uniform}(0, 1.6)$ for the standard deviations of the normally distributed risk aversion parameters. *Latent mixtures of utility models*: Finally, the modelling of latent mixtures of models via indicator variables, allows comparison between qualitatively different, as well as nested utility models, within one superordinate model[14]. The model indicator variable $z$ was set with non-informative uniform priors and was free to vary by subject. This represents our agnosticism toward which utility model is best under different dynamics. The posterior model probabilities (Fig 4C and 4D), estimated model frequencies (Fig 4E) and the protected exceedance probabilities (Fig 4F) were estimated via the Variational Bayesian Analysis toolbox[15] (mbb-team.github.io/VBA-toolbox/).

## Parameter estimation

Via a hierarchical model we estimated the posterior distribution of risk aversion parameters for a single dynamic-specific isoelastic utility model, given the choice data. This model is an

isoelastic model in which the risk aversion parameter is free to vary over dynamics, as well as over subjects. It is specified to be the same as the isoelastic utility model used in the model selection, except here the risk aversion parameter is estimated condition-wise, and there are no other utility models or latent model indicator variables.

## Results

### Gamble dynamics affect choice frequencies

Discrepant trials are the subset of trials in which a linear utility agent would choose a different gamble to a log utility agent (Fig 1E), 25 of 312 trials in the active session had this discrepant property. For example, in the Fig 1E, an agent with linear utility would be more likely to choose the left gamble, whereas an agent with logarithmic utility would be more likely to choose the right. By observing the choice proportions (CP) we obtain evidence about the dependency between choices and gamble dynamics (Fig 2A). We quantify evidence in terms of Bayes factors which are defined as the relative likelihood for one model over another, given the observation of the data. A Bayes factor of 10 for model 1 over model 2 indicates that the data is 10 times more likely given model 1 than given model 2. Levels of evidence are reported according to standard interpretations of Bayes factors (BF) [16,17]; ranging from anecdotal (1–3), moderate (3–10), strong (10–30), very strong (30–100) through to extreme (100>). We found moderate evidence against the hypothesis that subjects choose in favour of linear utility ($CP_{log} < 0.5$) under additive dynamics (Fig 2B, 2C, and 2D, $BF_{0-} = 3.678$, $M_{(CP)} = 0.4932$, SD = 0.1969, SEM = 0.04641, Bayesian central credibility interval: $BCI_{95\%}$ [0.395, 0.591], robust over prior widths). In contrast, we found extreme evidence for the hypothesis that subjects choose in favour of log utility ($CP_{log} > 0.5$) under multiplicative dynamics (Fig 2E, 2F, and 2G, $BF_{+0} = 460.4$, $M_{CP} = 0.718$, SD = 0.188, SEM = 0.044, $BCI_{95\%}$ [0.625, 0.812], robust over prior widths). Note that choosing in favour of linear utility ($CP_{lin} > 0.5$) is equivalent to choosing against logarithmic utility ($CP_{log} < 0.5$), and vice versa. The variable choice proportion for the multiplicative condition may not be normally distributed (Shapiro-Wilk p = 0.019), and thus we repeat the analysis with an equivalent non-parametric test (Wilcoxon Signed-ranks, V = 159, p < .001, effect size 0.86). Correspondingly, we found very strong evidence for the hypothesis of a within-subject increase in the choice proportions in favour of log utility when dynamics shift from additive to multiplicative (Fig 2H, 2I and 2K, $BF_{+0} = 52.38$, $M_{\Delta CP} = 0.225$, SD = 0.253, SEM = 0.060, $BCI_{95\%}$[0.099, 0.351], robust over prior widths). Finally, averaging across all models that entail possible combinations of factors and covariates, we found that the inclusion of the dynamic as a factor was uniquely favoured by the data (rmANOVA, $BF_{inclusion}$ = 80.2) with all other factors including order of testing showing $BF_{inclusion}$ < 1, see S2B Fig in S1 Text). Together, this shows strong evidence that in the discrepant trials, gamble dynamics exert a strong and systematic influence over choices.

### Estimates for utility model approximate time optimality

The model-free analysis of choice behaviour in the discrepant trials (25 per subject and condition) suggested that gamble dynamics affect choice behaviour in the direction predicted by time-optimality (Fig 2). As a next step, we fit an isoelastic utility model (also called constant relative risk-aversion utility function, CRRA) to the entire sample of choices (312 per subject and condition). For a discussion of testing the predictions of multiperiod utility as an alternative model, see the supplementary discussion in S3 Text. The isoelastic utility model has a single risk aversion parameter ($\eta$), negative values of which entail risk seeking, zero entails risk neutrality, and positive values entail risk aversion (Fig 3B. Eq 11). This model is suited to an explorative analysis of time optimality insofar as its parameter space contains values that are

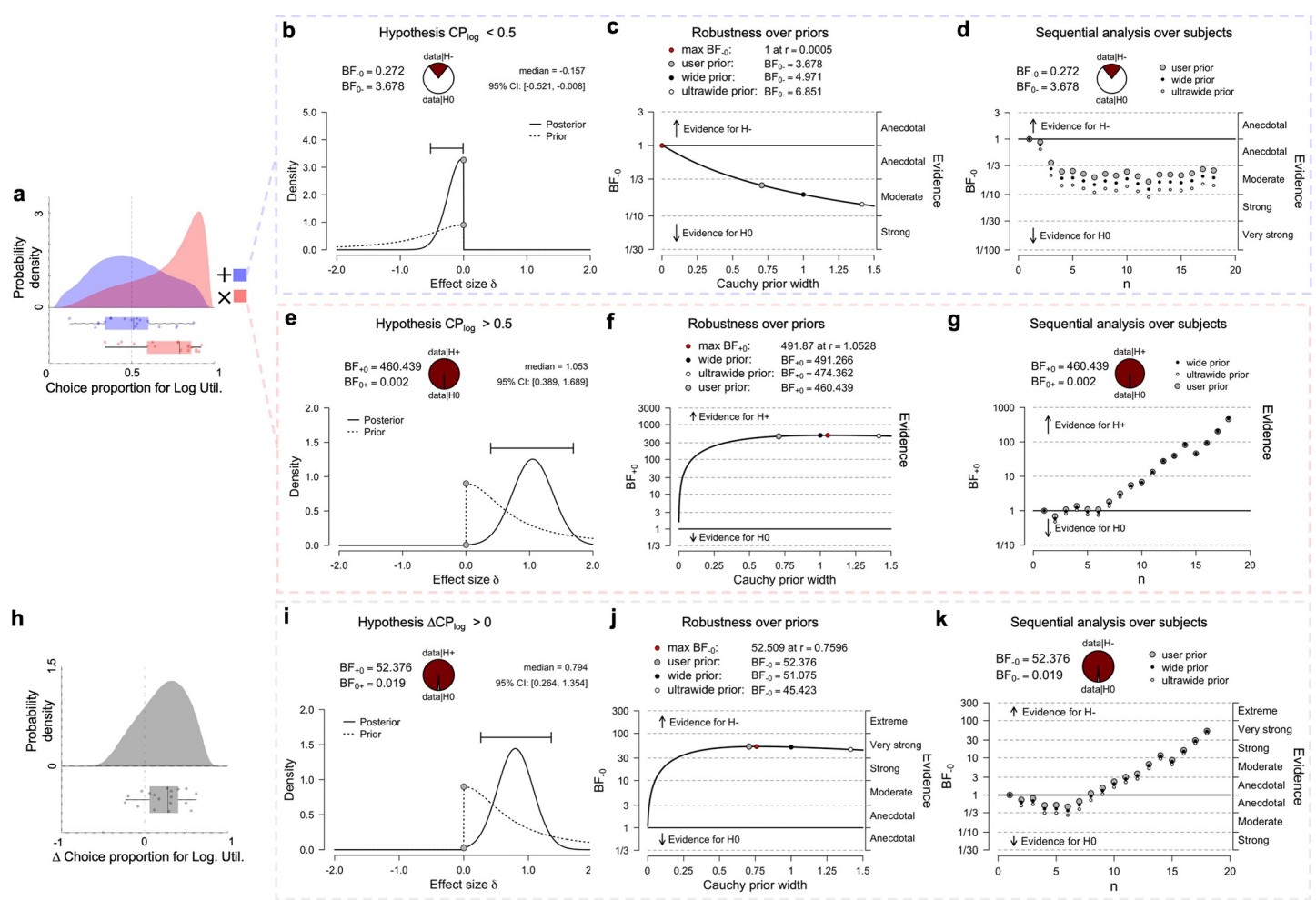

**Fig 2. Gamble dynamics affect choice frequencies. A,** raincloud plot[18] showing choice proportions in favour of log utility ($CP_{log}$), for multiplicative (red) and additive (blue) dynamic with split-half violin plot (top) and raw jittered data of individual subjects' choice proportions together with box and whisker plot (bottom). All box and whisker plots indicate range, 1st & 3rd quartiles, and median. **B,** prior and posterior density for the hypothesis that choice probabilities are in favour of linear utility ($CP_{log} < 0.5$) in terms of effect size, for the additive dynamic (Bayesian t-test), reporting Bayes factor in favour of $CP_{log}$ being lower than 0.5 (negative effect size, indicated by $BF_{-0}$) and its reciprocal in favour of the null hypothesis ($BF_{0-}$). **C,** robustness analysis of Bayes factors in B, showing that a less informative prior (ultrawide) would increase the Bayes factor in favour of the null hypothesis. **D,** sequential analysis showing how this Bayes factor changes with increasing numbers of subjects, with the different markers indicating different prior widths. **E** and **G,** equivalent analyses for the multiplicative dynamic for the hypothesis that choice probabilities are in favour of log utility ($CP_{log} > 0.5$). **H,** raincloud plot of the individual change in choice proportion ($\Delta CP_{log}$) where positive numbers indicate an increase under multiplicative dynamics. **I,** posterior and prior densities for the hypothesis that $CP_{log}$ is larger for multiplicative compared to additive dynamics (Bayesian Paired t-test). **J** and **K,** equivalent robustness and sequential analyses for this test.

time optimal solutions for both additive and multiplicative dynamics. Specifically, an agent that switches from risk neutrality with an $\eta$ of 0 under additive dynamics, and to risk aversion with an $\eta$ of 1 under multiplicative dynamics, is achieving time optimality by switching between linear and logarithmic utility. Thus, from this perspective, risk aversion should be calibrated to the dynamical setting to maximize the time average growth rate of wealth. Such time optimal agents would be expected to distribute their $\eta$ parameters around this optimal point as in Fig 3C (upper panel, where red and blue lines intersect), whereas agents with no systematic shift (dynamic-invariant agents), would distribute around the diagonal line (lower panel). In estimating a hierarchical Bayesian model of isoelastic utility (Fig 3A), we obtained separate posterior distributions of risk aversions for each gamble dynamic, which can be compared to

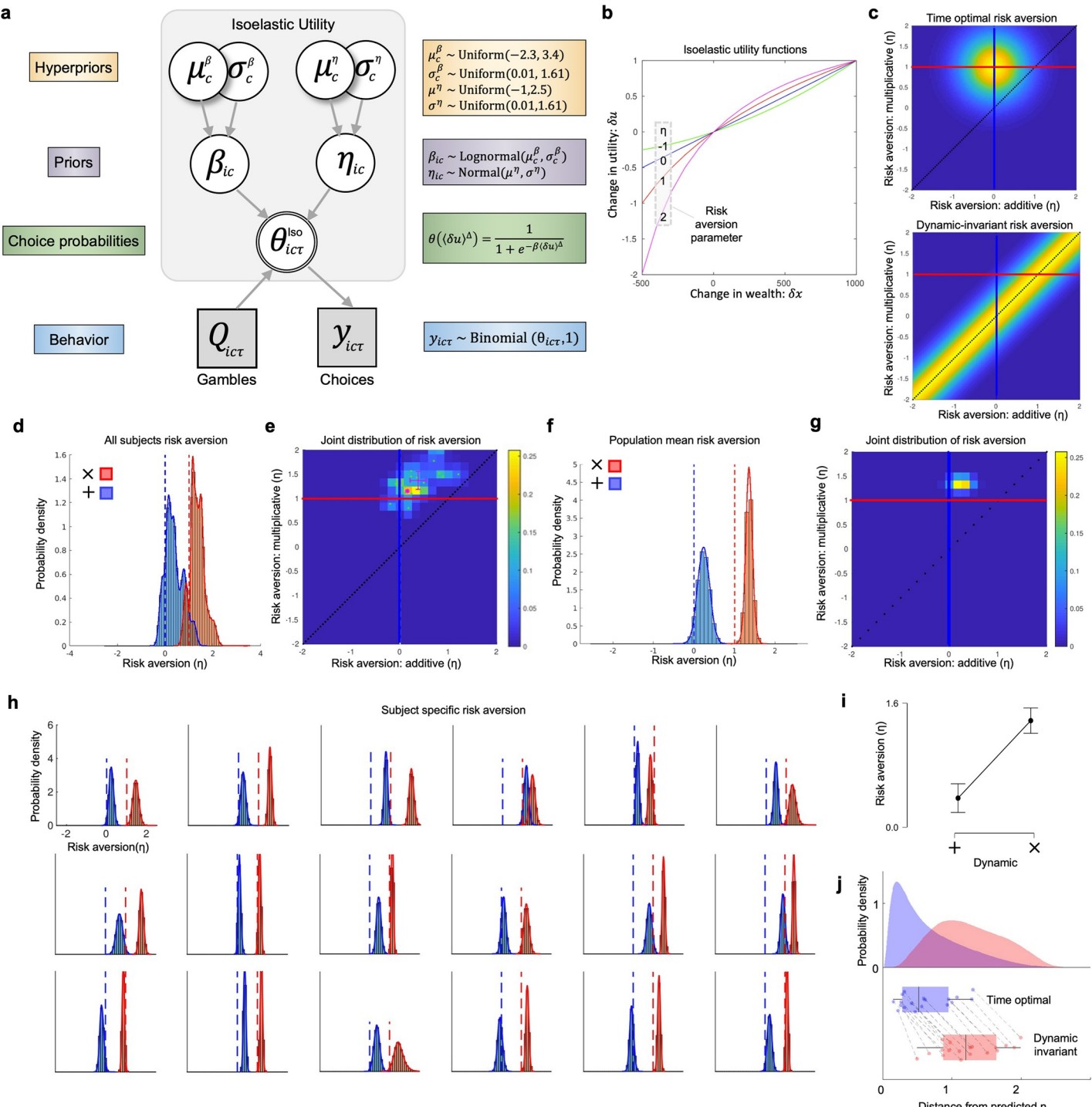

**Fig 3. Hierarchical Bayesian model for estimating dynamic-specific risk preference. A,** hierarchical Bayesian model for estimating risk preferences. Circular nodes denote continuous variables, square nodes discrete variables; shaded nodes denote observed variables, unshaded nodes unobserved variables; single bordered nodes denote deterministic variables, double bordered nodes stochastic variables. Along the left-hand side describes what role these variables play, and along the right side includes details on the distributions and logistic choice function. The data generating process (blue) which maps from theta to binary choice, is equivalent to a Bernoulli distribution. **B,** spectrum of utility functions entailed by different values of the risk aversion parameter η. **C,** schematic of model predictions of what values η will take for a time optimal (top) and dynamic-invariant isoelastic models (bottom). Heatmaps indicate probability density, with red and blue lines indicating time optimal risk aversion for additive and multiplicative conditions, respectively, intersecting at the time optimal strategy for both dynamics. Diagonal line indicates risk aversions that are invariant to dynamics. **D,** frequency distribution of risk aversion values collapsed over subjects for additive (blue) and multiplicative (red) dynamics. Dotted lines indicate time optimal values of risk aversion. **E,** joint distribution of dynamic-specific risk preferences. Maximum a posteriori (MAP) values are plotted for the group

(pink dot), and for each subject (cyan dots), and are superimposed over the group-level frequency distribution. Error-bars indicate the central $BCI_{95\%}$ for the subject-specific MAP values. Red, blue, and diagonal lines have same meaning as in panel C. **F** and **G**, same as D and E, but for posterior distributions of the population-level mean risk aversion. **H**, subject-specific risk aversion distributions, using same convention as D. **I**, mean risk aversion under each dynamic, bars show central $BCI_{95\%}$. **J**, raincloud plots of Euclidean distances of $MAP_\eta$ estimates to the predictions of the time optimal and dynamic invariant utility models. Grey lines link estimates from the same subjects.

these theoretical predictions. We refer to this as a dynamic-specific isoelastic model. Firstly, we find extreme evidence that risk aversion increases from additive to multiplicative dynamics ([Fig 3I], Paired t-test, $BF_{10} = 2.9 \times 10^7$, $M_\Delta = 1.001$, SD = 0.345, SE = 0.081, $BCI_{95\%}[0.829, 1.172]$), which is indistinguishable from the predicted size of change in $\eta$, under time optimality. As with the choice proportions, we found extreme evidence for the effect of gamble dynamic on risk aversion, compared to all other factors tested (rmANOVA $BF_{inclusion} = 2.45 \times 10^9$, all other factors $< 1$, S4B Fig in [S1 Text]). The same effect was evident when including covariates that account for differences in variance in wealth and wealth changes during the passive phase (rmANOVA $BF_{inclusion} = 1.19 \times 10^{10}$, all variance factors $< 1$, S4F Fig in [S1 Text]). Finally the frequency histograms of risk aversion marginalized over all subjects show that the maximum a posteriori (i.e. the most likely value of the posterior parameter distribution, $MAP_\eta$) value approximates the time optimal predictions for each dynamic: under additive dynamics, the distribution estimated from the data has a $MAP_\eta = 0.1506$, compared to the time optimal prediction of $\eta = 0$ ([Fig 3D], blue); under multiplicative dynamics, the distribution estimated from the data has a $MAP_\eta = 1.1534$, compared to the time optimal prediction of $\eta = 1$ ([Fig 3D], red). The joint distribution over risk aversion space ([Fig 3E]) shows that the MAP estimate of the joint distribution is likewise close to the optimal point indicated by the intersection of the prediction lines. A complementary visualisation of this correspondence comes from the posterior distribution of the population parameter for the mean of $\eta$ ([Fig 3F and 3G]). This indicates a qualitative agreement between the distribution of risk aversions, and the normative predictions of the time optimality model.

## Risk preferences are closer to predictions of time optimality

To test whether risk aversion values are explained better by time optimality ([Fig 3C] upper), or alternatively by a dynamic invariant utility model ([Fig 3C], lower), we computed the distance of each subject's risk aversion ($MAP_\eta$) to the predictions of each model. For the time optimal model this is the Euclidean distance to the time-optimal coordinate (0,1), and for the dynamic invariant model this is the distance to the closest point on the diagonal. We find extreme evidence that risk aversions are closer to the time optimal prediction ([Fig 3J], Paired t-test, $BF_{10} = 2.8 \times 10^{11}$, M = 0.623, $BCI_{95\%}$ [0.565, 0.681], S3E Fig in [S1 Text]), and that this is true for every subject tested. Together this shows that the time optimality model is a better predictor of risk aversion over different dynamics, than a null model which assumes no effect of dynamics on risk aversion.

## Order of gamble dynamics does not substantially affect choice

In the dynamic-specific isoelastic model, both the risk aversion parameter $\eta$ and the sensitivity parameter $\beta$ (modelling how sensitive choices are to differences in utility, [Eq 15]) are free to vary for each subject when the gamble dynamics change ([Fig 3A]). Plotting the joint distribution of both $\eta$ and $\beta$, affords visualisation of the effect of the dynamic on both risk aversion and on choice sensitivity (S5A Fig in [S1 Text]). We found that a switch from additive to multiplicative dynamics is associated with a characteristic shift in this parameter space toward greater risk aversion, and toward greater sensitivity. The order in which subjects experienced

different gamble dynamics was counterbalanced over subjects. In the subgroup that tested in the additive condition first (S5C Fig in S1 Text), the movement in parameter space is in the opposite direction to the subjects tested multiplicative condition first (S5B Fig in S1 Text), as predicted if the effect was primarily driven by the dynamic and not the order of testing. The inclusion probability for the order of testing had a Bayes Factor below one, indicating anecdotal evidence that the data disfavours its inclusion in the model ($BF_{inclusion}$ = 0.891, S4C Fig in S1 Text). Thus, there is no statistical evidence that the order of exposure to different gamble dynamics substantially affected choice.

## Deviation from time optimal value decreases time average growth rates for wealth

The relation between a subject's risk aversion and the time average growth rate of their choices (Eqs 8–9) can be noisy due to the probabilistic relation between utility and choice. This stochasticity is visible in the relation between the time average growth rates of the choices made and the risk aversion estimated for each subject under both dynamics, though the highest growth rates coincide with values close to the time optimal risk aversion (S6A Fig in S1 Text). Further, we found that the closer the subjects shifted their risk aversion toward time optimal values, the higher the time average growth rates of their wealth, given their choices for both additive (S6B Fig in S1 Text, $\tau$ = -0.428, $BF_{10}$ = 10.51, $BCI_{95\%}$ [-0.655, -0.086]) and multiplicative dynamics (S6C Fig in S1 Text, $\tau$ = -0.502, $BF_{10}$ = 30.88, $BCI_{95\%}$ [-0.711, -0.131]). Thus, the risk aversion parameter that best describes a subject's choices is predictive of their time average growth rate. This illustrates that deviating from time optimality has substantial negative consequences for growing wealth, as implied by theory. These are economically meaningful differences in growth rates on the timescale of the game itself.

## Bayesian model selection supports time optimality over other utility models

The dynamic-specific isoelastic utility model suggested that subjects dynamically adapt their choice behaviour in a way predicted by time optimality. We next compared the predictive adequacy of three models, an isoelastic model, a prospect theory model (Eq 10), and the time optimal model (Eq 12), detailed in Fig 4A and 4B. The time optimal model is fixed in its theoretical predictions for the population means of $\eta$, restricted to be 0 for additive dynamics and 1 for multiplicative dynamics. However, the variance around this mean is a free parameter to account for the plausible assumption that not all subjects are phenotypically identical. Prospect theory has two utility parameters whose means are not fixed at the population level but are free to vary within standard restrictions that define the theory (See Models section). Finally, the isoelastic model has one utility parameter that is estimated across both sessions, whose mean is free to vary at the population level. Markov chain Monte Carlo sampling of this model results in posterior frequencies for the model indicator variable $z$ that are interpreted as posterior probabilities for each model, estimated for each subject[14]. Most subjects had most of their probability mass located over the time optimal model (Fig 4C), as is evident from the marginal probability over subjects (Fig 4D). Computing protected exceedance probabilities, which measure how likely it is that any given model is more frequent (estimated frequencies in Fig 4E) than all other models in the comparison set, we found that the time optimal model had an exceedance probability of 0.976 (Fig 4F) which corresponds to very strong evidence for it being the most frequent ($BF_{Time-PT}$ = 76.9, $BF_{Time-Iso}$ 80.6).

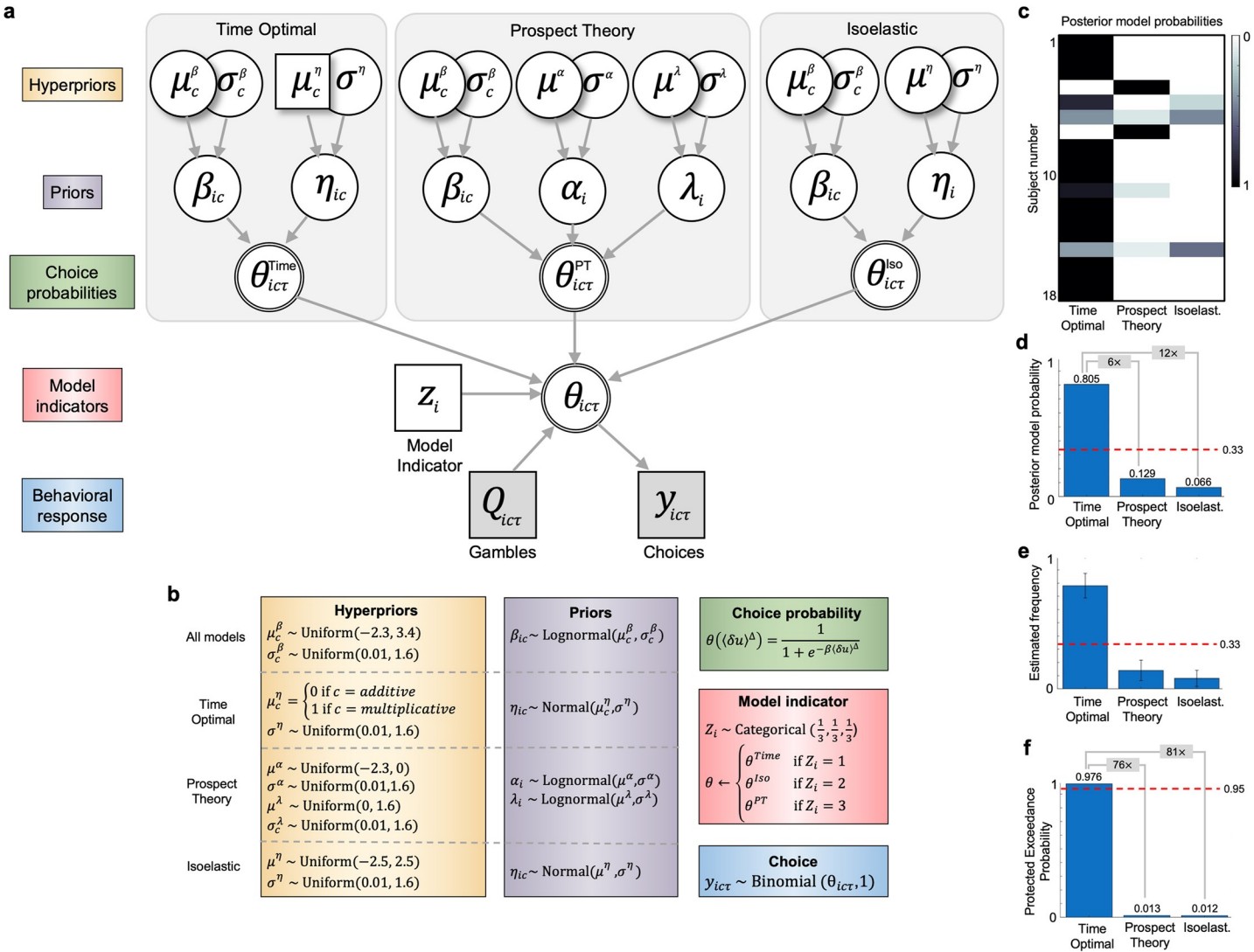

**Fig 4. Bayesian hierarchical latent mixture model and model selection results. A,** graphical model according to conventions of Fig 3. This model adds a model indicator variable (z) to modelling latent mixtures of the three different utility models nested within it. Note that for prospect theory risk preference parameter $\alpha$, there is one parameter for gains, and another for losses. **B,** hyperprior and prior distributions, including structural equations, choice functions, and choice generating distributions. Hyperpriors for $\alpha$ are duplicated to model gains and losses separately. **C,** posterior model probabilities for each model based on the model indicator variables representing each utility model. **D,** posterior model probabilities summed over subjects, with the red bar indicating prior probabilities assuming equal prior probability for the three utility models. **E,** estimated model frequencies from the cohort and error bars as standard deviations. **F,** protected exceedance probabilities for each utility model being the most frequent.

## Discussion

### Summary

By manipulating the dynamical properties of simple gambles, we show that ergodicity-breaking can exert strong and systematic effects on risk-taking behavior. Switching from additive to multiplicative gamble dynamics reliably increased risk aversion, which in most subjects tracked close to the levels that maximize the time average growth of their in-game wealth (wealth, hence). We show that these effects are well predicted by a model of time optimality based on ergodic theory and cannot be adequately explained by the prevailing models of utility in economics and psychology.

## Main findings

The time optimal model assumes that agents prefer their wealth to grow faster, and that this preference for faster growth is stable. From these assumptions, it can be shown that to maximize the time average growth rate of wealth, agents should adapt their utility functions according to the wealth dynamics they face, such that changes in utility are rendered ergodic[3]. From this, a few simple predictions can be derived, each increasing in specificity. First, to approximate time optimal behavior, different gamble dynamics require different ergodicity mappings. Thus, when an agent faces a different dynamic, this should evoke the observation of a different utility function. This was observed, in that all subjects showed substantial changes in their estimated utility functions (Fig 3H). Second, in shifting from additive to multiplicative dynamics, agents should become more risk averse. This was also observed in all subjects. Third, the predicted increase in risk aversion should be, in the dimensionless units of relative risk aversion, a step change of +1. The mean step change observed across the group was +1.001 (BCI$_{95\%}$ [0.829,1.172]). Fourth, to a first approximation, most (not all) participants modulated their utility functions from ~linear utility under additive dynamics, to ~logarithmic utility under multiplicative dynamics (Fig 3D). Each of these utility functions are provably optimal for growing wealth under the dynamical setting they adapted to, and in this sense they are reflective of an approximation to time optimal behavior. Finally, model comparison revealed strong evidence for the time optimal model compared to both prospect theory and isoelastic utility models, respectively. The latter two models provide no explanation or prediction for how risk preferences should change when gamble dynamics change, and even formally preclude the possibility of maximising the time average growth rate when gamble dynamics do change. In line with this explanatory gap, both prospect theory and isoelastic utility models were inadequate in predicting the choices of most participants (Fig 4C).

## Logarithmic utility

Maximising the expected value of the logarithm of wealth is an established strategy in finance, where it is deployed as part of the Kelly criterion, a formula for sizing bets that leads almost surely to higher wealth compared to any other strategy, in the long run[19]. Though used in practice, it has not been empirically demonstrated whether subjects can naively intuit this strategy. The results of this experiment speak to the plausibility of this idea, insofar as they demonstrate that, at least in the subjects tested here, utility functions under multiplicative dynamics approximated logarithmic utility. Our main result, however, is not simply that logarithmic utility is a good approximation of behavior under multiplicative dynamics, but rather that risk preferences change from ~log utility to ~linear utility, as predicted if agents are maximising time average growth. In effect, ergodic theory can be thought of as providing a generalization of the Kelly criterion by generalising it to a larger class of optimisation behaviors, only two of which we test here. To verify that log utility is time optimal for multiplicative dynamics in this experimental design, we refer the reader to Fig 1G, which shows that with repetitions of this exact experiment, wealth grows fastest for the agent with log utility. The equivalent demonstration for additive dynamics and linear utility is evident in Fig 1F.

## Differences between conditions

In the passive phase of the experiment, in which subjects learnt the effects of the stimuli on their wealth, there were different dynamics at play in the different conditions that could in principle lead to differences in the experience of the subject. One such difference is in the variance of the wealth changes, which were higher on average for the multiplicative than the additive condition. These variances were variable across participants; however, we found no

evidence for them explaining the observed differences in risk aversion (S4F Fig in S1 Text). It is possible that other features of the wealth trajectories may differ between the two conditions, which is unavoidable due to the two dynamics being qualitatively different. Due to the fact that wealth levels needed to be bounded between 0 and 5000kr during the passive phase, many of the possible wealth trajectories had to be discarded when sampled during the experimental setup, prior to each subject's session. For the multiplicative condition, substantially more trajectories with positive excursions were discarded, whereas for the additive conditions more negative excursions were discarded. While this skews the representativeness of the random process generating the stimuli and makes it non-independent, it should be noted that this is irrelevant to the gambles of the decision-making phase which are drawn from a qualitatively different generative process. Furthermore, the selective filtering imposed by this process acted to attenuate the differences in the wealth trajectories between conditions. Another consideration is whether subjects learnt the stimuli better under the additive condition. Such an effect should be apparent in the distribution of risk aversion parameters, in which greater uncertainty should manifest in less precise posterior distributions. This was not observed (Fig 3F). To date, there has been one partial replication of this study, focusing on the fidelity with which subjects can discriminate the value of these stimuli, finding that there is no strong evidence for a difference between conditions, even when stimuli are learnt in one fifth of the time[20].

## Statistical considerations

The size of the cohort (achieved n = 18) was constrained to concentrate power within subjects, and by the high-stakes design, in which each participant could walk away with up to 750 USD in payout, a total that would typically approach or exceed their monthly disposable income. Restricting our inferences to this cohort, the effect was consistent across all participants, and was reproducible across different inferential approaches. In general, the strength of the evidence we obtained from individuals likely derives from the fact that the game is high stakes, and from the fact that we collected a large number of decisions ($> 600$ per participant) over a large space of distinct gambles (320 per participant). This affords opportunity for stringent testing, even between utility models that may make overlapping predictions. The strength of the evidence thus observed, likely derives from this being a large and consistent effect size, that was likely driven by the large incentives, a fundamental shift in strategy caused by the dynamics, and by the large number of trials. Indeed, for many of the tests conducted, high degrees of evidence are obtained from sample sizes substantially lower than the full group sample. The fact that discrimination between utility models is possible under our framework is evident from its ability to recover parameters and model identities from synthetically generated agents (S7 Fig in S1 Text). Finally, a cohort with a sample size of 18 may lead to concerns about the experiment being underpowered. We note however that interpreting statistical evidence in terms of power once the data is already collected can be misleading[21]. The statistical evidence we report is in the form of Bayes factors, which represent the relative predictive performance of the models and hypotheses. If the low sample size for the cohort were a cause of low predictive adequacy in the models, then this would already be reflected in the Bayes factors, which they are not.

## Novelty, validity, and generalisation

This experiment is, to our knowledge, the first empirical test of an ergodic theory of decision making, and it presents the first data suggesting that risk preferences can systematically change as a function of changing dynamics. We introduce a new paradigm of dynamical conditioning in which outcomes under different dynamics are passively learnt, and then actively chosen. It

will therefore be important to generalize time optimal behaviors to more conventional paradigms, where dynamics and outcomes might be more explicitly presented, and where existing dynamical models may have a better chance of being deployed. We deliberately chose not to reveal the outcome of the gamble decisions in the active session to avoid choices being biased by previous outcomes. This is different from many real-life decisions where one can immediately observe the outcomes of one's choices. We also apply a relatively novel modelling method (the hierarchical latent mixture model), to our knowledge not previously applied to decision making paradigms. Furthermore, the paradigm was relatively novel in terms of its very high levels of incentives, and the relatively large numbers of decisions made per subject. It will therefore be important to investigate whether the proposed phenomena generalize to lower incentives, and whether it can be inferred with lower volumes of data. The experiment here tests only two different dynamics, though other dynamics predict other time optimal utility functions. A more stringent test of the ergodic theory will require generalization to other dynamics not tested here. The ethical constraint of not allowing subjects to lose money at the end of the experiment potentially impacts more on the additive condition, since negative wealth is impossible under multiplication of positive growth factors. Strictly speaking, the prediction that linear utility is the time optimal utility function for additive dynamics assumes only additive dynamics without any such constraints. The fact that the data are reasonably well explained by a theory which ignores these constraints suggests that, to a first approximation, these constraints are not critical for predicting the behavior of these participants. In general, we are careful not to make formal claims about the generalisation of this time optimality behavior beyond the subjects tested, and beyond the paradigm used. In essence, we make the limited claim that these results suggest that time optimal behavior is possible in at least some subjects (all tested so far) under certain specific laboratory conditions. Establishing time optimality as a more general behavioral phenomenon will require multi-centre replication, and then broader generalisation to ascertain its robustness to paradigm variations. Finally, we note that the time optimal behaviour observed here was specific to the in-game wealth that was endowed to subjects at the start of the experiment. We make no claims about time optimal decision making for real wealth outside of the laboratory, other than that this suggests it is at least cognitively plausible. Decision making about real wealth may be subject to more complex dynamics than those tested here, involving multiple interacting variables, and over longer time horizons. Thus, testing the generalisation of time optimal decision making to real wealth contexts will require careful interrogation.

## Theoretical considerations

The dependency between dynamics and risk aversion that we observe here is relevant to a widespread assumption that utility functions are stable over time[5,22,23,24]. Primarily, this is motivated on epistemological grounds. If utility is to predict behaviour in future settings, then it must be stable, otherwise if behavior changes, it is not known if this is due to a change of setting or preference, or both[23]. However, this is contradicted by multiperiod utility models [25], as well as a diversity of empirical demonstrations of preference instability. In animals, including humans, there is evidence suggesting that risk preferences depend on homeostatic' [26–30], circadian[31], and affective states[32]. Test-retest stability in the same settings, though typically reported as modest[33], can be relatively high when estimated using hierarchical models of the sort used here[34]. The findings reported here place the stability of utility in a broader context by connecting to an optimality framework for how utility functions should change in response to changes in one's environmental dynamics. This casts the dynamical dependence of utility functions observed here, not as preference instability per se, but simply

as a manifestation of a stable preference for growing wealth over time when facing different circumstances. We refer the interested reader to a theoretical critique[35] written in response to a preprint of this paper and our reply (S4 Text).

## Final remarks

Models of decision-making are predominantly developed without recourse to dynamical considerations and are typically tested in settings that implicitly evoke additive dynamics. The theories developed under these conditions are then assumed to generalize to settings in which additive dynamics may no longer apply, and multiplicative dynamics likely dominate, which may contribute to the predictive inadequacy of many models. This initial evidence motivates a need to develop and further scrutinize theories of decision making that are explicitly conditioned on ergodic foundations.

## Supporting information

**S1 Text. Supplementary Results and Modelling. S1 Fig in S1 Text**. **Growth rates for outcomes and gambles. A**, on Day$^\times$ (upper table) a growth factor is the factor by which current wealth is multiplied when a given stimulus is encountered in the passive session. The effect of each stimulus can thus be expressed as a multiplicative growth rate (in units of *growth factor per trial*). Computing the natural logarithm of the growth factor per trial gives a continuous growth rate (in units of *% change per trial*). On Day$^+$ (lower table), growth increments are the additive amounts by which wealth changes, and thus the growth rate is an additive growth rate (in units of *DKK per trial*). **B,** each gamble is comprised of two different possible outcomes, here denoted in terms of pairs of stimuli. Each cell shows the time average growth rate associated with each gamble in the space of possible gambles. The cells with red text indicate the 16 gambles that were presented in the active sessions. For Day$^\times$ (upper) the time average multiplicative growth rates of each gamble have units *% change per trial*. For Day$^+$ (lower) the time average additive growth rates have the units of *DKK per trial*. **S2 Fig in S1 Text**. **Model comparisons and analysis of effects for choice proportions. A,** the table of model probabilities, Bayes factors and error terms, for a repeated measures ANOVA on the choice proportions for discrepant trials. The meanings of each column are described in the text. **B,** the inclusion probabilities for all factors of interest across all models, along with the Bayes factors for their inclusion. **S3 Fig in S1 Text**. **Descriptive statistics, priors & posteriors of hypothesis test, robustness tests and sequential analyses. A-D,** effect of dynamics on changing risk aversion parameters. **H-K,** comparison of the deviation of each model predictions of risk aversion parameters to those observed. **L-O,** effect of deviating from time optimality on the time average growth rates of subjects' choices, under additive dynamics. **P-S,** equivalent effect under multiplicative dynamics. **S4 Fig in S1 Text**. **Tables and no-brainers. A,** descriptive statistics for the risk aversion parameter, η. SD -standard deviation; SE—standard error mean. **B,** table of models compared in repeated measures ANOVA for risk aversions. Column headings described in main text. **C,** table of Analysis of effects shows the Bayes factors for the inclusion of each factor across all of the models in b. P(incl) indicates the prior probability of each effect across all models. BF$_{Inclusion}$ is the Bayes factor for the inclusion of that factor, comparing all models with vs. without that factor. **D,** table for Kendall correlation, where correlations between time averages (ta) and distances from optimal risk aversion parameters ('distfromopt') are tabulated. **E,** proportion of correct responses (dominant chosen) in the no-brainer trials. Red line indicates chance performance. **F,** Analysis of effects shows Bayes factors for the inclusion of each factor, including both the dynamic, and the covariates derived from the variances in wealth and changes in wealth. **S5 Fig in S1 Text**. **Displacements of model**

**parameters as a function of dynamics. A,** displacement in parameter space caused by changing the gamble dynamic. Filled and empty circles indicate additive and multiplicative dynamics, respectively. **B-C,** equivalent displacements splitting subjects according to the temporal order of their experience of the dynamics. **S6 Fig in S1 Text. Time average growth rates as a function of risk aversion parameters. A,** distribution of subject specific time average growth rates and risk aversion under both dynamics. **B,** correlation between time average additive growth rate of subject's choices and deviation of subject's risk aversion away from the time optimal value. **C,** equivalent plot for multiplicative dynamics. **S7 Fig in S1 Text. Parameter and Model recovery. A**, parameter recovery for several populations of synthetic agents with different combinations of η parameters. Each panel shows the posterior η distribution, marginalized over subjects, estimated via the same model and code as the real data shown in Fig 3. **B**, model recovery for three different groups of synthetic agents, for the three models compared. Color range shows posterior model probabilities as in Fig 4C (black = 1, white = 0). Posterior model probabilities map strongly onto the ground truth model identities, insofar as the first nine agents were synthesized via a time optimal model, the next nine from a prospect theory model, and the final nine from an isoelastic utility model. Note that the isoelastic agent obtains small posterior model probabilities in the range of ~0.05 for the two other models, for only 2/9 parameter values.
(DOCX)

**S2 Text. Synthetic agents.**
(DOCX)

**S3 Text. Supplementary Discussion. S1 Fig in S3 Text. State-space graphs for different versions of the task.** A, state-space graph of a simplified version of the game comprising only two possible gamble pairs, and with only two trials. B, the same state-space for four trials. C, The state-space for this experiment for a single trial. In all graphs A, B and C-, red dots indicate branches between possible pairs of gambles, yellow dots indicate possible branches between choices, and green dots indicate branches between different possible outcomes.
(DOCX)

**S4 Text. Reply to Doctor, Wakker, and Wang.**
(DOCX)

**S5 Text. Experimental checklist.**
(DOCX)

**S6 Text. Subject Instructions.**
(DOCX)

## Acknowledgments

We thank Ole Peters, Alex Adamou, Yonatan Berman, Mark Kirstein, Tobias Andersen, Jason Collins, Peter Dayan, Brad Cameron, Chris Merrill, Adam Goldstein, Alex Imas, Ilari Lehti, Peter Wakker and Sven Resnjanskij for helpful discussions. Thank you to Félix Hubert for making S1 Fig in S3 Text.

## Author Contributions

**Conceptualization:** David Meder, Finn Rabe, Tobias Morville, Kristoffer H. Madsen, Hartwig R. Siebner, Oliver J. Hulme.

**Data curation:** Finn Rabe, Oliver J. Hulme.

**Formal analysis:** Kristoffer H. Madsen, Oliver J. Hulme.

**Funding acquisition:** Hartwig R. Siebner.

**Investigation:** David Meder, Finn Rabe, Oliver J. Hulme.

**Methodology:** David Meder, Kristoffer H. Madsen, Oliver J. Hulme.

**Project administration:** David Meder, Finn Rabe, Hartwig R. Siebner, Oliver J. Hulme.

**Resources:** Hartwig R. Siebner.

**Software:** Kristoffer H. Madsen, Magnus T. Koudahl, Oliver J. Hulme.

**Supervision:** David Meder, Hartwig R. Siebner, Oliver J. Hulme.

**Validation:** Oliver J. Hulme.

**Visualization:** Oliver J. Hulme.

**Writing – original draft:** David Meder, Oliver J. Hulme.

**Writing – review & editing:** David Meder, Finn Rabe, Tobias Morville, Kristoffer H. Madsen, Magnus T. Koudahl, Ray J. Dolan, Hartwig R. Siebner, Oliver J. Hulme.

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
