## [Decision Letter · Decision Letter 0]

14 Dec 2020

Dear Hulme,

Thank you very much for submitting your manuscript "Ergodicity-breaking reveals time optimal decision making in humans" for consideration at PLOS Computational Biology.

As with all papers reviewed by the journal, your manuscript was reviewed by members of the editorial board and by several independent reviewers. In light of the reviews (below this email), we would like to invite the resubmission of a significantly-revised version that takes into account the reviewers' comments.

Two of the reviewers were very enthusiastic about the paper, and one of them (R2) raised several important criticisms. It is of particular importance to address R2's concerns about experimental design and sample size.

We cannot make any decision about publication until we have seen the revised manuscript and your response to the reviewers' comments. Your revised manuscript is also likely to be sent to reviewers for further evaluation.

Sincerely,

Samuel J. Gershman

Deputy Editor

PLOS Computational Biology

Reviewer's Responses to Questions

**Comments to the Authors:**

Reviewer #1: This paper demonstrates that (a) people’s decisions about risky monetary gambles are sensitive to the dynamic by which this money grows/shrinks (additive vs. multiplicative); and (b) this sensitive behavior closely approximates the normative benchmarks to maximize an arithmetic mean value for additive dynamics and a geometric mean value for geometric dynamics. The authors convincingly show that a model whose utility matches this normative benchmark depending on the dynamic vastly outperforms more standard utility models that do not consider dynamics, such as prospect theory.

I have been incredibly impressed by this manuscript since it was released as a preprint a year or so ago, and this is one of the rare cases where I do not have substantive comments about the paper’s quality. Even without the complex Bayesian modeling, the paper makes a very compelling case that multiplicative dynamics dramatically change people’s risk preferences in an optimal way — a point that seems to have been largely ignored in the decision-making literature that I am aware of. (Perhaps this has been studied in some detail in the financial literature, as relates to the Kelly criterion, but this is not my expertise.)

My only (minor) worry is about the paper’s comprehensibility. Specifically, the figures are quite detailed and often include a lot of detail about the specifics of the Bayesian model fits that were used. Although this is useful content to include for people who are familiar with this style of modeling, it seems like some of it would be more appropriate for the supplement. For example, many of the robustness checks in Fig. 2 seem like they would fit better in the supplement in order to put more emphasis on the really critical subpanels like (a). Similarly, Fig. 3 requires a huge amount of effort to process, and I’m not so sure panels like (i) and (j), which look at order effects, need to be in the main text, given that order did not seem to play a major role on behavior. This would help put emphasis on the critical subplots like (c), which show how much better the time-optimal model performs relative to a model with static risk aversion. Critically, I believe the main findings of this paper should be comprehensible to people who don’t have a strong background in Bayesian modeling. Even without the Bayesian analyses, it should be quite clear from the figures that dynamics matter in the predicted ways.

Reviewer #2: The paper presents a behavioral experiment, where subjects first passively learn values of fractals (which could be additively or multiplicatively increasing their endowment – representing the two main conditions of the experiment) and then make forced binary choices between 50/50 gambles that include those fractals. At the end of the active phase, 10 gambles are selected at random and applied to their initial endowment.

The paper argues that when dealing with multiplicative gambles, individuals tend to use log utility, while when dealing with additive increases, they behave as if they employ linear utility. These results are supported by a series of Bayesian statistical analyses and model fits using various utility models.

While I do not question the results of the experiment in general (although see power considerations below), and I think it’s carefully implemented (and the model fitting exercise is impressive), I unfortunately (1) do not believe it is sufficiently novel to justify publication in PLOS Computational Biology; (2) do not see how the results of this specific experiment contribute to our understanding of economic behavior in dynamic settings. I wish I could be more positive, but I think the situation would be different if the neuroimaging results were presented as well.

- First and foremost, the novelty of the *ergodicity* considerations (as well as the works of Ole Peters) is strongly disputed by economists. Maximizing the expected value of the logarithm of wealth is known in the finance literature since 1950s as the Kelly criterion and has been studied in numerous publications on portfolio optimization (note though that the use of log utility does not guarantee success in all cases, and the expected utility theory still can be a viable alternative). The macroeconomic and asset pricing literatures both use log utility in a wide arrange of models. The standard expected utility theory does not require ergodicity and has little to do with it. That being said, the laboratory evidence on dynamic gambling behavior and log utility is quite limited (for a rather weak example, see Haghani and Dewey (2016), Rational Decision-Making under Uncertainty: Observed Betting Patterns on a Biased Coin), so this might be a potentially fruitful line research. However:

- I cannot see how the dynamics are really represented in the experimental design of this study. In the passive task, the dynamics are quite clear: subjects observe their wealth change in real time, going up and down, in order to learn the values associated with the fractals. In the active task however, there is no real dynamics – as far as I understand. The subjects make choices between gambles without any feedback, so they pretty much make these choices completely independently. At the end of the task, 10 random choices are selected (this number seems important, but is never justified), gambles are realized, and the results are applied to their initial wealth level, all at the same time (sequentially, in random order? - again, if I understand the methods correctly). How does this correspond to any dynamic optimality considerations as presented in Figure 1f-g (also see a minor point below – does the optimality effect really require the time horizon of weeks and wealth of millions of crowns? Does it mean that the real effect of using log utility in the task is economically insignificant?)? It seems that when subjects are making a choice in each specific trial, they do not know the initial wealth level these choices would be applied to (since, I assume, all 10 choices are applied sequentially at the end, with the wealth endowment as the initial value)? To summarize the argument, the task design does not seem to capture the idea it is supposedly aimed to demonstrate: that people, when making decisions in a *multi-period* environment with gambles applied multiplicatively to their wealth, do that optimally (using log utilities, or the Kelly criterion). Yes, the experiment shows that log utility functions fit better in the multiplicative case. However, it is not at all clear that log utility is optimal specifically for this experimental setup (this needs to be clearly shown for the specific gambles used in the task, and all possible initial wealth levels), and how these results generalize to a truly dynamic setting.

- The study is clearly underpowered. The main result is the difference between a choice proportion of 0.49 (for the additive case) and a choice proportion of 0.71 (for the multiplicative case), with SD being around 0.2 in both cases. I understand and appreciate the Bayesian exercise, as well as the model fits, but in the end, everything just comes down to these two proportions. And if one does the power calculations for these numbers, they do not look great. Power concerns are only mentioned once in the Methods section: “The sample number was based on general guidelines for the minimal number of subjects required for medium effect sizes in neuroimaging datasets”. But this is not a neuroimaging study, this is a behavioral study. Being honest, I do not really understand why this is being published separately from the neuroimaging results since it would be a stronger and more interesting paper if those were combined. In 2020, publishing a behavioral effect based on 18 subjects with 320 trials each (with only 25 trials actually measuring the effect) is not really up to the current standards of replicability (even though the analyses were preregistered).

Minor remarks:

- Prospect theory *is not* a utility model; it’s not a well-defined utility function, it’s a value function. This mix-up happens throughout the manuscript.

- I am not sure if Figure 1e has any value since the reader does not know what these fractals correspond to.

- Figure 1f-g is not clear: the colors are difficult to tell apart, a legend could be useful. For figure 1g – do I understand correctly that the effect is only clearly visible for wealth levels of 10^100 crowns? That doesn’t make much empirical sense. How do these calculations look like when they are applied to the stakes used in the experiment?

- End of the introduction: “impose a consistent effect on utility functions” – would be better to specify here what the effect is and why it is assumed that the utility functions are causally affected by the experimental manipulation.

Reviewer #3: In this paper, the authors present a model-driven analysis of decision making under risk in two types of scenarios: additive outcomes where wealth is increased or decreased by a particular amount, or multiplicative outcomes where wealth is increased or decreased by a particular proportion. The authors show – through choice proportions, parameter estimates, and formal model comparisons via model indicator variable – that participants adjust their decision strategies across these two conditions, corresponding to a time-optimal utility model.

In general, I found the methods and analyses thorough. The Bayesian analyses were carried out in a rigorous way by checking robustness to different priors, reporting both the Bayes factors and credibility intervals, and performing comparisons on both the raw data and the models. Unusually, I didn’t really see any missing analyses or conceptual flaws in the paper, so most of my comments will center on clarification and style rather than pertain to the substantive conclusions.

I found the results and discussion convincing, but I think the exposition in the introduction could be more comprehensive. I struggled to understand from the single example how time average growth rate is calculated, and why a logarithmic utility function is time optimal. Going through the calculations and the predictions of different utility functions as they relate to dynamic changes in wealth would be helpful.

Another change that might help clarify the content of the paper is if the model descriptions were provided in the introduction, as opposed to materials & methods. To an extent, the order of the sections for the journal is working against the authors here – my interpretation of the ordering is that materials & methods should be details needed to replicate exactly the experiment and results in the paper, but not details that are necessarily critical to understanding the results themselves. The authors may have to work around this by re-arranging the paper to an extent. Reading linearly through the paper, I had no idea what delta-u meant, or where nu was included in the models, or exactly what the distinction between isoelastic utility and time optimal utility models were. These are of course all provided in the materials & methods section, but it would be better suited as part of the intro – if only to avoid having to constantly flip back and forth between the results and methods. In addition to being critical to understanding the results, setting up the model predictions in the intro should also help motivate the experimental and analysis approaches (particularly the additive-multiplicative conflict trials), helping the reader understand why the findings are interesting.

**Have all data underlying the figures and results presented in the manuscript been provided?**

Reviewer #1: Yes

Reviewer #2: Yes

Reviewer #3: Yes

PLOS authors have the option to publish the peer review history of their article (what does this mean?). If published, this will include your full peer review and any attached files.

Reviewer #1: No

Reviewer #2: No

Reviewer #3: No
---

## [Decision Letter · Decision Letter 1]

8 Feb 2021

Dear Hulme,

Thank you very much for submitting your manuscript "Ergodicity-breaking reveals time optimal decision making in humans" for consideration at PLOS Computational Biology.

As with all papers reviewed by the journal, your manuscript was reviewed by members of the editorial board and by several independent reviewers. In light of the reviews (below this email), we would like to invite the resubmission of a significantly-revised version that takes into account the reviewers' comments. The main issue at hand is whether you can substantively address (in the paper, not just the response letter) the theoretical concerns about ergodicity arguments referenced below, as they pertain to your experimental design.

We cannot make any decision about publication until we have seen the revised manuscript and your response to the reviewers' comments. Your revised manuscript is also likely to be sent to reviewers for further evaluation.

Sincerely,

Samuel J. Gershman

Deputy Editor

PLOS Computational Biology

Reviewer's Responses to Questions

**Comments to the Authors:**

Reviewer #2: Interesting that the authors chose to "talk" themselves out in response to virtually all of my comments instead of making (sometimes perhaps just small and/or clarifying) changes to the paper, even in the cases where they are objectively wrong (such as calling the prospect theory a utility function).

Just a few days after I submitted my initial review, Nature Physics published a response to Ole Peters' paper, written by several economists:

Doctor, Jason N., Peter P. Wakker, and Tong V. Wang. "Economists’ views on the ergodicity problem." Nature Physics 16.12 (2020): 1168-1168.

This paper also provides a series of very strong criticisms for the paper we are discussing now (see Appendix A in the Supplements), and I must suggest that the authors should address these before the paper is published (although I understand it might be impossible with the current design). Some of them mirror my comments (perhaps in a more convincing and clear way).

To be more specific, when the authors respond "When realising at the end of the experiment, it makes no difference whether they are applied all at once e.g. wealth + (a + b +... n), or sequentially (wealth +a, then that new wealth +b...), since you get the same answer.", this is simply not true. The order *does* matter if the outcomes are realized sequentially, and this is the whole point of investigating dynamic decision making. The Doctor et al. response raises the same concern; this is the single major flaw of the design.

Finally, if the claimed novelty includes "The first ever experimental test of ergodicity economics, itself a nascent branch of economics", I would strongly suggest to submit the paper to an economics journal, if the authors still wish to receive feedback from a qualified audience most familiar with the subject.

**Have all data underlying the figures and results presented in the manuscript been provided?**

Reviewer #2: Yes

PLOS authors have the option to publish the peer review history of their article (what does this mean?). If published, this will include your full peer review and any attached files.

Reviewer #2: No
---

## [Decision Letter · Decision Letter 2]

13 May 2021

Dear Hulme,

Thank you very much for submitting your manuscript "Ergodicity-breaking reveals time optimal decision making in humans" for consideration at PLOS Computational Biology. As with all papers reviewed by the journal, your manuscript was reviewed by members of the editorial board and by several independent reviewers. The reviewers appreciated the attention to an important topic. Based on the reviews, we are likely to accept this manuscript for publication, providing that you modify the manuscript according to the review recommendations.

As you will see from the review below, your paper is unlikely to satisfy everyone. However, my view is that this paper makes a valuable contribution and should be published, even if it has some limitations. The final important task for you is to address these limitations clearly, acknowledging in particular the issues raised by the review below. I want to make sure that this paper is not read as over-claiming the generality of the conclusions, especially concerning the translation from these laboratory experiments to real-world decisions.

Sincerely,

Samuel J. Gershman

Deputy Editor

PLOS Computational Biology

[LINK]

Reviewer's Responses to Questions

**Comments to the Authors:**

Reviewer #2: I appreciate that the authors made the effort to edit the manuscript in response to my previous comments. I have read their new response as well as the response to Doctor et al.

My new comment will be more of a methodological discussion rather than a direct comment on the details of the paper. I do completely understand the experiment that was run, and I do not really have any issues with the procedure. I would say my issue with the paper lays primarily in the interpretation of the results and implied importance of the findings. I find it difficult to provide a specific recommendation, but I hope that my thoughts could be useful for the editorial decision – and could help to understand why there is such a backlash against this line of research among economists.

There is a reason economists dislike laboratory experiments (if we do not count those who run lab experiments themselves). In many cases, the findings have very little external validity. It is a tired criticism, but it is sometimes still appropriate. Specifically, it is very difficult to run a macroeconomic experiment, or an experiment that deals with individual “wealth”. Real decisions over wealth take days, months, and sometimes years. The stakes are extremely high for the individual that makes these decisions (it’s their wealth after all). These stakes are often present in macroeconomic models (e.g. see Epstein-Zin preferences), but very few try to test them in the lab – and if they do, they provide very little actual value to the community. The reasons are obvious: the “wealth” in the lab is not actual wealth, it’s pennies; the decisions do not matter for the subject’s economic well-being in the long run; the decisions are not made with the same level of carefulness as the real-life decisions.

When we test a simple risk choice task in the lab, there are many real-life decisions that could correspond to the task; even buying a lottery ticket. When the paper claims that subjects dynamically estimate their “wealth” trajectory, that terminology is just not appropriate. There are no “wealth” decisions to be made, the subjects simply do a series of extremely fast choices about some pocket money… without even knowing their current “wealth” (i.e. endowment) level.

From the supplements:

“This is because the subjects did not know the following five details necessary to perform a backward induction from terminal wealths: a) how many gambles they would face; b) what gambles they will face; c) the outcome of each gamble; d) which choices would be realised; e) what their current wealth is at the time of choosing.”

So the subjects did not even know their current “wealth” at the time of choosing. I still fail to see how this reflects a “dynamical” decision-making process. It seems that the subjects were acting on extremely limited information; something that they would not do in real life, when making choices that determine their future wealth.

From the response:

“It would matter to decision makers if the outcomes were realised during the game, and they were revealed to the subject at the time of their occurence, because this would update their knowledge of their current wealth. Clearly changing the order changes the sequence of wealths within the game, which could plausibly change the pattern of decisions made. Maybe that is the crux of our disagrement?”

I see that the authors realize themselves that that observing individual outcomes could be important to the subjects – as it is to decision-makers in real life. I can hardly imagine a scenario where this information would not be available to a household. I can’t see the reason why this was not made an explicit feature of the experimental design. But again, an ideal experiment that would test the ergodicity theory would use much higher sums of money (approaching the monthly income of the subjects); longer time horizons; choices that do not take seconds, but at least hours.

I hope that my point is clear. I do not think that this experiment tests exactly what the authors are hoping it should test (decisions over wealth); it opens a potentially interesting topic, but it is unlikely make an impact for the economic literature due to these methodological concerns.

I think it would be beneficial for the manuscript to discuss the limitations of the experiment (and its relation to real-life decisions or external validity) in greater detail, but ultimately it is up to the editor to decide whether the paper makes a valuable contribution. My impression is that unfortunately the experiment itself (as it was done) does very little to increase our knowledge regarding how dynamic economic decisions are made. However, it might generate an interesting discussion in the literature – so there might be some value in that!

**Have the authors made all data and (if applicable) computational code underlying the findings in their manuscript fully available?**

Reviewer #2: Yes

PLOS authors have the option to publish the peer review history of their article (what does this mean?). If published, this will include your full peer review and any attached files.

Reviewer #2: No

Figure Files:

Data Requirements:

Reproducibility:

References:

---

## [Editor Report · Decision Letter 3]

28 Jun 2021

Dear Hulme,

We are pleased to inform you that your manuscript 'Ergodicity-breaking reveals time optimal decision making in humans' has been provisionally accepted for publication in PLOS Computational Biology.

Best regards,

Samuel J. Gershman

Deputy Editor

PLOS Computational Biology

---

## [Editor Report · Acceptance letter]

3 Sep 2021

PCOMPBIOL-D-20-02113R3 

Ergodicity-breaking reveals time optimal decision making in humans

Dear Dr Hulme,

I am pleased to inform you that your manuscript has been formally accepted for publication in PLOS Computational Biology. Your manuscript is now with our production department and you will be notified of the publication date in due course.

With kind regards,

Livia Horvath
